# Conserved collateral antibiotic susceptibility networks in diverse clinical strains of *Escherichia coli*

Nicole L. Podnecky[1], Elizabeth G.A. Fredheim[1], Julia Kloos[1], Vidar Sørum[1], Raul Primicerio[1], Adam P. Roberts[2,3], Daniel E. Rozen[4], Ørjan Samuelsen [1,5] & Pål J. Johnsen[1]

There is urgent need to develop novel treatment strategies to reduce antimicrobial resistance. Collateral sensitivity (CS), where resistance to one antimicrobial increases susceptibility to other drugs, might enable selection against resistance during treatment. However, the success of this approach would depend on the conservation of CS networks across genetically diverse bacterial strains. Here, we examine CS conservation across diverse *Escherichia coli* strains isolated from urinary tract infections. We determine collateral susceptibilities of mutants resistant to relevant antimicrobials against 16 antibiotics. Multivariate statistical analyses show that resistance mechanisms, in particular efflux-related mutations, as well as the relative fitness of resistant strains, are principal contributors to collateral responses. Moreover, collateral responses shift the mutant selection window, suggesting that CS-informed therapies may affect evolutionary trajectories of antimicrobial resistance. Our data allow optimism for CS-informed therapy and further suggest that rapid detection of resistance mechanisms is important to accurately predict collateral responses.

[1] Department of Pharmacy, Faculty of Health Sciences, UiT The Arctic University of Norway, 9037 Tromsø, Norway. [2] Department of Parasitology, Liverpool School of Tropical Medicine, Pembroke Place, Liverpool, L3 5QA, UK. [3] Research Centre for Drugs and Diagnostics, Liverpool School of Tropical Medicine, Pembroke Place, Liverpool, L3 5QA, UK. [4] Institute of Biology, Leiden University, Sylviusweg 72, PO Box 9505, 2300 RA Leiden, The Netherlands. [5] Norwegian National Advisory Unit on Detection of Antimicrobial Resistance, Department of Microbiology and Infection Control, University Hospital of North Norway, 9037 Tromsø, Norway. Correspondence and requests for materials should be addressed to N.L.P. (email: nicole.podnecky@uit.no) or to P.J.J. (email: paal.johnsen@uit.no)

The evolution and increasing prevalence of antimicrobial resistance is driven by the consumption and misuse of antimicrobials in human medicine, agriculture, and the environment[1–3]. Historically, the threat of antimicrobial resistance was overcome by using novel antimicrobials with unique drug targets. However, the discovery rate of new antimicrobial agents has dwindled[4–6] and severely lags behind the rate of resistance evolution[7]. While concerted scientific, corporate, and political focus is needed to recover antimicrobial pipelines[8–10], there is an urgent need for alternative strategies that prolong the efficacy of existing antimicrobials and prevent or slow the emergence, spread, and persistence of antimicrobial resistance. Current global efforts to improve antimicrobial stewardship largely focus on reducing overall antimicrobial consumption and increasing awareness of resistance development[9,11–13]. While these efforts will affect the evolution and spread of resistance, mounting evidence suggests that these changes alone will not lead to large-scale reductions in the occurrence of antimicrobial resistance[14–18].

Several recent studies have examined novel treatment strategies using multiple antimicrobials that could reduce the rate of resistance emergence and even reverse pre-existing resistance. These approaches, collectively termed selection inversion strategies, refer to cases where resistance becomes costly in the presence of other antimicrobial agents[19]. Among the most promising of these strategies are those based on a phenomenon first reported in 1952, termed collateral sensitivity (CS), where resistance to one antimicrobial simultaneously increases the susceptibility to another[20]. CS and its inverse, cross-resistance (CR), have been demonstrated for several bacterial species and across different classes of antimicrobials[21–27]. These results have formed the basis of proposed CS-informed antimicrobial strategies that combine drug pairs[22,28] or alter temporal administration, e.g. drug cycling[21,29]. CS-informed strategies would force bacteria to evolve resistance along a predictable trajectory, resulting in CS; this predictability could be exploited to ultimately reverse resistance and prevent the fixation of resistance and multi-drug resistance development at the population level of bacterial communities.

Initial in vitro experiments support using CS-based strategies to re-sensitize resistant strains[21] and reduce rates of resistance development[29]; however, the broader application of this principle depends on predictable bacterial responses during antimicrobial therapy. This predictability must be general for a given drug class and should not vary across strains of the same species. To date, most studies of CS and CR have focused on describing collateral networks[21–23] using resistant mutants derived from single laboratory-adapted strains and limited numbers of clinical isolates. Two studies on *Pseudomonas aeruginosa* have investigated CS in collections of clinical isolates[30,31]. However, these studies lack either baseline controls[30] or sufficient genetic diversity among tested strains[31]. As valuable as earlier work has been, the responses of single strains (laboratory or clinical) may not be representative of CS and CR responses in other strains.

To address this limitation, here we focus on understanding collateral networks in clinical urinary tract isolates of *Escherichia coli* with selected resistance to drugs widely used for the treatment of urinary tract infections: ciprofloxacin, trimethoprim, nitrofurantoin, and mecillinam. We investigate collateral networks to 16 antimicrobials from diverse drug classes in 10 genetically diverse clinical strains (corresponding to 49 laboratory-generated mutants) to assess the factors contributing to collateral responses (both CS and CR). This approach allows us to identify variation in the sign and magnitude of collateral responses and identify mechanisms of CS and CR that are preserved in various genetic backgrounds. Using multivariate statistical modeling, we show

that resistance mutations, particularly those affecting efflux pumps, and the relative fitness of resistant isolates are more important determinants of collateral networks than genetic background. Our results support the idea that collateral responses may be predictable.

## Results

**Collateral responses vary between and across resistance groups.** We examined collateral responses to antimicrobial resistance in a panel of 10 genetically diverse (Supplementary Fig. 1a–b) *E. coli* strains isolated from urinary tract infections. For each of these pan-susceptible strains (Supplementary Fig. 1c)[32], a single resistant mutant was generated to each of four individual antimicrobials used to treat urinary tract infections: ciprofloxacin, trimethoprim, nitrofurantoin, and mecillinam. Here we define resistance group as the collection of mutants from the 10 different genetic backgrounds that were selected for resistance to the same antimicrobial. Mutants resistant to mecillinam required only a single selection step, while multiple selection steps were required to select for resistance above clinical breakpoints for the remaining antimicrobials. In total, 40 resistant mutants were generated with resistance levels above clinical breakpoints, as determined by antimicrobial susceptibility testing using both gradient strip diffusion (Supplementary Table 1) and inhibitory concentration 90% ($IC_{90}$)[21] testing (Table 1). The two methods are correlated, but $IC_{90}$ measurements allow for more robust detection of small relative differences in susceptibility[33,34]. Changes in the $IC_{90}$ of resistant mutants from each respective wild-type strain (Supplementary Fig. 2) were compared for 16 antimicrobials (Table 2). Overall, collateral responses were observed in 39% (233/590) of possible instances (Supplementary Table 2); of these 49% (115/233) were associated with only a 1.5-fold change in $IC_{90}$. Such small changes would not be observed by typical two-fold antimicrobial susceptibility testing methods frequently used in clinical laboratories.

Overall CR was more frequent than CS, 141 versus 92 instances (Supplementary Table 2), and collateral networks varied considerably between resistance groups. We observed 19 cases of conserved collateral responses (Fig. 1a), where CR or CS to a specific antimicrobial was found in ≥50% of the mutants within a resistance group, defined as $CR_{50}$ or $CS_{50}$, respectively. For each $CR_{50}$ and $CS_{50}$ observation, $IC_{90}$ results were further assessed by generating dose–response curves of representative strain:drug combinations (Supplementary Fig. 3). Inhibition of growth was shown to vary across antimicrobial concentrations between resistant mutants and respective wild-type strains, confirming the changes in antimicrobial susceptibility determined by the $IC_{90}$ assays.

During the selection of resistant mutants, we often observed colonies of varying size for all resistance groups, suggesting changes to bacterial fitness. To test this, we measured the growth rates of mutants relative to the respective wild-type strains (Supplementary Fig. 4). In general, mutants resistant to ciprofloxacin and mecillinam displayed severely reduced growth rates, suggesting high costs of resistance. Relative growth rates varied between 0.34–0.75 with a mean of 0.53 for ciprofloxacin-resistant mutants and between 0.49–0.79 with a mean of 0.64 for mecillinam-resistant mutants. Mutants resistant to nitrofurantoin and trimethoprim displayed lower fitness effects, and several resistant mutants harbored apparent cost-free resistance mutations (Supplementary Fig. 4). Only two of ten nitrofurantoin-resistant mutants and four of ten trimethoprim-resistant mutants displayed an apparent cost of resistance. Relative growth rates varied between 0.93–1.05 and 0.68–1.07 with averages of

**Table 1 Description of *Escherichia coli* strains used in the study and average IC$_{90}$ changes following antimicrobial selection**

| Strain | ST[a] | Origin | CIP[b] | | MEC[b] | | NIT[b] | | TMP[b] | |
|---|---|---|---|---|---|---|---|---|---|---|
| | | | WT[c] | [c]CIP[R] | WT[c] | [c]MEC[R] | WT[c] | [c]NIT[R] | WT[c] | [c]TMP[R] |
| K56-2 | 73 | Greece | 0.014 | 16 | 0.146 | >30 | 8 | >64 | 0.225 | >28 |
| K56-12 | 104 | Portugal | 0.016 | 1.67 | 0.273 | 28 | 7.33 | >64 | 0.563 | 32 |
| K56-16[d] | 127 | Portugal | 0.009 | 3 | 0.167 | 18.7 | 4 | >64 | 0.25 | >30 |
| K56-41 | 73 | Greece | 0.016 | 2.33 | 0.104 | 13.3 | 6 | >64 | 0.25 | 6.67 |
| K56-44[d] | 12 | Greece | 0.013 | 1.67 | 0.141 | 16 | 6.67 | >64 | 0.375 | 6 |
| K56-50 | 100 | Greece | 0.012 | 3 | 0.141 | 10.7 | 12 | >64 | 0.172 | 18 |
| K56-68 | 95 | Sweden | 0.014 | 4 | 0.141 | 30 | 6.67 | >64 | 0.208 | 18.7 |
| K56-70 | 537 | Sweden | 0.007 | 2.67 | 0.083 | >32 | 4.67 | >64 | 0.25 | 14.7 |
| K56-75[e] | 69 | UK | 0.008 | 1.17 | 0.063 | 13 | 6 | >64 | 0.167 | 5.33 |
| K56-78 | 1235 | UK | 0.015 | 6 | 0.141 | 16 | 8 | >64 | 0.5 | 7.33 |

[a]Multi-locus sequence type (ST)
[b]The average IC$_{90}$ values (μg mL$^{-1}$) of three or more biological replicates for wild type (WT) and resistant ($^R$) mutants to ciprofloxacin (CIP), mecillinam (MEC), nitrofurantoin (NIT), and trimethoprim (TMP). Individual results above detection limits (MEC = 32 μg mL$^{-1}$, NIT = 64 μg mL$^{-1}$, TMP = 32 μg mL$^{-1}$) were analyzed as those values, yielding final results with uncertainty (>average). EUCAST Clinical Breakpoints v 7.1 for Enterobacteriaceae[63] were: >0.5 μg mL$^{-1}$ CIP, >8 μg mL$^{-1}$ MEC, >64 μg mL$^{-1}$ NIT, and >4 μg mL$^{-1}$ TMP
[c]The strain number names the WT, and designations CIP$^R$, MEC$^R$, NIT$^R$, and TMP$^R$ describe which drug the isolates were selected with, and resistance achieved
[d, e]Strains containing the Col156 or Col(MP18) replicon, respectively

**Table 2 List of antimicrobials used in this study**

| Antimicrobial[a] | Abbreviation | Drug class | Drug target(s) | Solvent |
|---|---|---|---|---|
| Amoxicillin | AMX | β-lactam (Penicillin) | Cell wall synthesis | Phosphate buffer[b] |
| Azithromycin | AZT | Macrolide | Protein synthesis (50S) | ≥95% Ethanol |
| Ceftazidime | CAZ | β-lactam (Cephalosporin) | Cell wall synthesis | Water + 10% (w w$^{-1}$) Na$_2$CO$_3$ |
| Chloramphenicol | CHL | Amphenicol | Protein synthesis (50S) | ≥95% Ethanol |
| Ciprofloxacin | CIP | Fluoroquinolone | DNA replication, cell division | 0.1 N HCl |
| Colistin | COL | Polymyxin | Cell wall & cell membrane | Water |
| Ertapenem | ETP | β-lactam (Carbapenem) | Cell wall synthesis | Water |
| Fosfomycin | FOS | Phosphonic | Cell wall synthesis (MurA) | Water |
| Gentamicin | GEN | Aminoglycoside | Protein synthesis (30S) | Water |
| Mecillinam | MEC | β-lactam (Penicillin) | Cell wall synthesis (PBP2) | Water |
| Nitrofurantoin | NIT | Nitrofuran | Multiple[c] | Dimethyl sulfoxide |
| Trimethoprim | TMP | Antifolate | Folate synthesis (FolA) | Dimethyl sulfoxide |
| Sulfamethoxazole | SMX | Antifolate | Folate synthesis (FolP) | Dimethyl sulfoxide |
| TMP + SMX (1:19) | SXT | Antifolate | Folate synthesis (FolA + FolP) | Dimethyl sulfoxide |
| Temocillin | TEM | β-lactam (Penicillin) | Cell wall synthesis | Water |
| Tetracycline | TET | Tetracycline | Protein synthesis (30S) | Water |
| Tigecycline | TGC | Tetracycline | Protein synthesis (30S) | Water |

[a]When available, final antimicrobial concentration was determined using manufacturer-provided or calculated drug potencies, otherwise potency was assumed to be 100%. Aliquots were stored at −20 or −80 °C in single-use vials. All antimicrobials and chemical solvents were obtained from Sigma-Aldrich (St. Louis, MO, USA) with the exception of ciprofloxacin (Biochemika, now Sigma-Aldrich) and temocillin (Negaban®)
[b]0.1 mol L$^{-1}$, pH 6.0 phosphate buffer supplemented with 6.5% (v v$^{-1}$) 1 M NaOH (sodium hydroxide)
[c]Nitrofurantoin is thought to target macromolecules including DNA and ribosomal proteins, affecting multiple cellular processes, including protein, DNA, RNA, and cell wall synthesis

0.99–0.94 for nitrofurantoin- resistant and trimethoprim-resistant mutants, respectively.

**Ciprofloxacin resistance linked to conserved collateral responses**. Nearly half (108/233, 46%) of the observed collateral responses were in ciprofloxacin-resistant mutants, while the remaining 125 were distributed between the other three resistance groups (Supplementary Table 2). Within the ciprofloxacin-resistant group, the majority of collateral responses were CR (70/108, 65%). Additionally, CS responses in ciprofloxacin-resistant mutants were the most conserved in our dataset, with CS to gentamicin occurring in 8 of 10 strains and CS to fosfomycin in 7 of 10 strains (Fig. 1a). Gentamicin and other aminoglycosides are important for the treatment of a wide range of infections[35], while fosfomycin is primarily used for treatment of uncomplicated urinary tract infections[36,37]. The ciprofloxacin-resistant mutants were also unique in the magnitude of observed

changes, with cases of CR close to 30-fold and CS as high as six-fold changes in IC$_{90}$ (Supplementary Fig. 2).

**Characterization of antimicrobial-resistant mutants**. We hypothesized that CS and CR variation in and between resistance groups could be attributed to different mutations causing resistance in each strain. Using whole genome sequencing, we identified a total of 149 mutations in the resistant mutants (Supplementary Data 1–4). Of these, 88 mutations affect previously described or putative antimicrobial resistance-associated genes, gene-regions, or pathways (Supplementary Data 1–4). The remaining mutations were found in other cellular processes not known to affect antimicrobial susceptibility (e.g. metabolic pathways and virulence factors), such as mutation to the FimE regulator of FimA that was frequently observed in mecillinam-resistant mutants (Supplementary Data 2). Aside from FimE, we did not observe mutations in regions unrelated to resistance

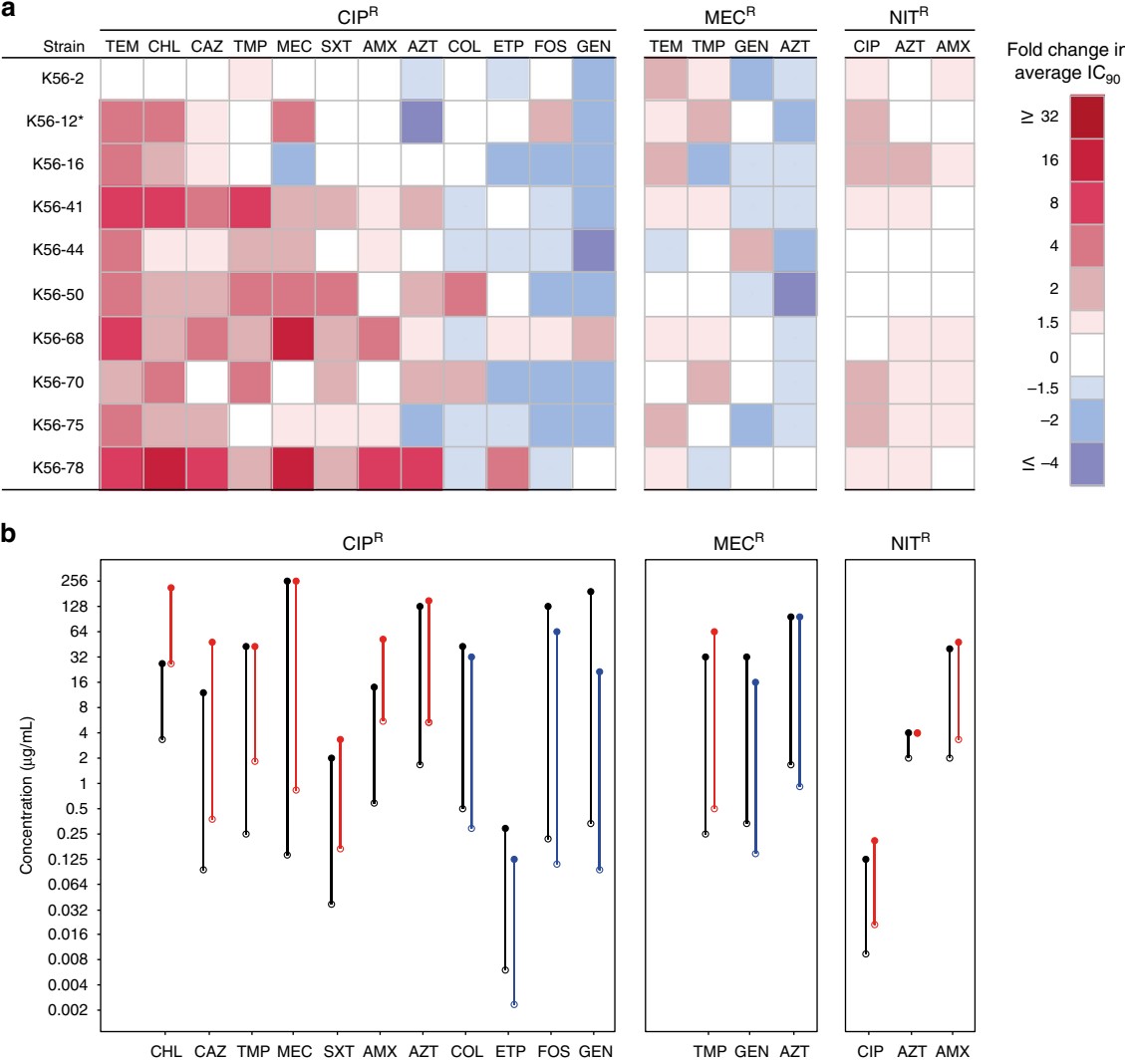

**Fig. 1** Conserved collateral responses in antimicrobial resistant mutants. **a** Relative change in antimicrobial susceptibility was determined by comparing average $IC_{90}$ values of resistant mutants to the respective wild-type strain. Collateral responses that were found in ≥50% of the strains are displayed, excluding CR observed in all trimethoprim-resistant mutants to trimethoprim-sulfamethoxazole (see Supplementary Fig. 2). Antimicrobials are ordered by most frequent CR (red; left) to most frequent CS (blue; right) for each group. *The slow growing K56-12 $CIP^R$ was incubated an additional 24 h for $IC_{90}$ determination. **b** The average $IC_{90}$ (open circles) and average mutation prevention concentration (MPC; filled circles) were determined and compared between resistant mutants (colored) with collateral responses, either CS (blue) or CR (red), and their respective wild-type strain (black) in strain:drug combinations representing conserved collateral responses, excluding temocillin. The mutant selection window (vertical lines) was defined as the range between $IC_{90}$ (lower bound) and MPC (upper bound). K56-16 $NIT^R$ had equivalent $IC_{90}$ and MPC values for azithromycin, thus no mutation selection window was reported. Generally, changes in MPC values reflected observed $IC_{90}$ changes, shifting the mutation selection window upwards or downwards accordingly. In 8/10 tested combinations an increase in $IC_{90}$ value (CR) from wild-type to resistant mutant correlated with at least a small increased MPC, with the remaining combinations showing no change in MPC value. Similarly, decreased $IC_{90}$ values (CS) correlated with decreased MPCs (5/7)

across mutants of the same resistance group (parallel evolution), suggesting that such mutations had limited, if any, effect on collateral responses in this study.

For each of the 40 resistant mutants at least one putative resistance mechanism was identified, including mutations to previously described antimicrobial drug targets and promoters of drug targets, drug-modifying (activating) enzymes, regulators of efflux pumps, RNA polymerases and mutations to other metabolic and biochemical processes that may contribute to resistance (Table 3). Briefly, all but one ciprofloxacin-resistant mutant contained mutations in both *gyrA* and efflux regulatory genes and/or gene-regions likely affecting efflux expression (*acrAB* and/or *mdtK*), while one strain had only drug target mutations and displayed the well-described GyrA (S83L) and

ParC (G78D) mutation combination (Supplementary Data 1). Both efflux and drug target mutations are frequently found in surveys of clinical isolates[38–41]. Nitrofurantoin-resistant mutants had mutations in one or both nitro-reductases (*nfsA*, *nfsB*) and the majority of strains had additional mutations in *mprA*, which encodes an efflux regulator of EmrAB-TolC pump expression (Supplementary Data 3). Mutants resistant to trimethoprim contained mutations either in *folA* and/or its promoter or genetic amplification of a large region containing *folA* (Supplementary Data 4). The mecillinam-resistant mutants are unique in that they evolved as single step mutants, where a single mutation could confer clinical resistance to mecillinam. Resistance development for the remaining three drugs required several steps, as multiple mutations were required for resistance above clinical breakpoints.

**Table 3 The number of antimicrobial resistant mutants with resistance-associated mutations**

| Resistance mechanism | | CIP$^R$ | MEC$^R$ | NIT$^R$ | TMP$^R$ |
|---|---|---|---|---|---|
| Drug target | Modification | 10$^a$ | | | 6 |
| | Overproduction | | | | 6 |
| Drug activation | Nitroreductase disruption | | | 10 | |
| Drug uptake | Porin mutation | 1 | | | |
| Efflux | AcrAB-TolC | 7 | | 1 | |
| | MdtK | 9 | | 1 | |
| | MdfA | | | 1 | |
| | EmrAB-TolC | | | 7 | |
| | ABC transport | | 1 | | |
| ppGpp synthesis (stringent response activation) | Stringent response | | 4 | | |
| | tRNA synthesis | | 4 | | |
| | tRNA processing | | 1 | | |
| | Cellular metabolism | | 3 | | |

$^a$All mutants resistant to ciprofloxacin contained one mutation in the *gyrA* gene, except the K56-2 CIP$^R$ mutant that contained two mutations in *gyrA* and a mutation in *parC*

In total, 12 different mutations in genes and/or cellular processes previously linked to mecillinam resistance were identified in this resistance group (Supplementary Data 2)[42].

The ciprofloxacin-resistant group displayed a clear trend where conserved CR responses were strongly linked to mutations in efflux regulatory regions suggesting that *gyrA* drug target mutations had a limited effect on CS and CR. Trimethoprim-resistant mutants also had few collateral responses, likely due to the specific mechanism of resistance affecting a single unique drug target (i.e. overexpression/alteration of FolA). To further investigate the effects of drug target mutations, we assessed the collateral responses of mutants generated following a single selection-step with ciprofloxacin. These first-step mutants contained single, non-synonymous mutations to *gyrA* and no other mutations (e.g. in efflux pumps) linked to ciprofloxacin resistance (Supplementary Data 1). The IC$_{90}$ of these strains was uniformly lower than in ciprofloxacin-resistant strains containing multiple resistance mutations. Few collateral responses were observed in these first-step mutants (Fig. 2), and none were conserved across different strain backgrounds. These results suggest that most collateral responses observed in the ciprofloxacin-resistant mutants are due to the observed efflux mutations.

**Efflux and fitness are main contributors to collateral responses**. Multivariate statistical approaches were used to investigate the extent to which genetic (strain) background, resistance group, the putative mechanism of resistance (in particular efflux-related mutations), growth rate, and the fitness cost of resistance explain the variation in collateral responses. All factors were investigated individually (Supplementary Fig. 5a–e). Throughout the remaining analyses we focus mainly on efflux-related mutations, rather than resistance group, to explicitly address putative mechanisms of resistance, and relative fitness rather than growth rate.

We estimated several models with individual, or a combination of, factors to assess their effect size and significance given some level of collinearity between fitness and efflux-type (Fig. 3, Supplementary Fig. 5a–r). A model including strain background, relative fitness, and efflux-related mutations as factors explained 62.5% of the total variation in IC$_{90}$ values (Fig. 3a, b, Supplementary Table 3). In this three-factor model there was clear separation of the mutants by resistance group (Fig. 3a). The ciprofloxacin-resistant mutants showed strong CR towards temocillin, chloramphenicol, ceftazidime, and amoxicillin, separating this resistance group from the others along the first ordination axis (Fig. 3a, b). Along the second ordination axis, mecillinam-resistant isolates were distinct, had CR to temocillin, and were more likely to have CS towards drugs, such as azithromycin and chloramphenicol (Fig. 3a, b). Both efflux-type and relative fitness were significant predictors when tested alone and in combination (Supplementary Table 3). The model (Fig. 3a, b) also revealed that strain background had a non-significant ($p = 0.993$) contribution (Supplementary Table 3). Even when modeled alone (Supplementary Fig. 5a), strain background only accounted for 6.5% of the variation and was non-significant (Supplementary Table 3).

We initially hypothesized that genetic background would significantly affect collateral responses. Our initial analysis suggests that it does not. Arguably, the inclusion of IC$_{90}$ data from the drugs to which primary resistance was selected could confound the analysis, despite our efforts to minimize these effects using log-transformed data. We used the same approaches to assess a subset of collateral responses, excluding data for all of the 40 resistant mutants to five antimicrobials containing the drugs used for selection (ciprofloxacin, mecillinam, nitrofurantoin, trimethoprim) and trimethoprim-sulfamethoxazole. Within the subset model, patterns consistent with the full model were observed, but with a lower degree of clustering by resistance group (Fig. 3c). For example, K56-2 CIP$^R$ is now co-localized with the mecillinam-resistant isolates, indicating that this isolate is distinct from other ciprofloxacin-resistant mutants (Fig. 3c), which still showed strong tendencies of CR to temocillin, chloramphenicol, ceftazidime, and amoxicillin (Fig. 3c, d). Despite these changes, efflux-type and fitness were still significant predictors of collateral networks, and strain background remained non-significant (Supplementary Table 3) when modeled alone (Supplementary Fig. 5f) and in two-factor combinations (Supplementary Fig. 5n–o), but had a limited, significant contribution ($p = 0.040$), determined by permutation tests, in the three-factor model (Fig. 3c, d, Supplementary Table 3). However, mutations in efflux-related genes and gene regulators were the strongest predictor of collateral responses tested, explaining over 33% of the variation in the subset. Fitness alone also had significant predictive value, but to a lesser extent (17% variation explained). It is important to note that we observed a correlation between efflux mutations and relative fitness that is likely explained by reduced fitness resulting from the cost of over-expression of efflux pump(s)[39].

To investigate the influence of resistance mechanism on IC$_{90}$ variation at a higher resolution, we modeled each resistance group separately relating the putative resistance mechanism (beyond efflux-type) and fitness separately and in combination (Supplementary Fig. 6a–o). However, potentially due to a lower number of samples within each resistance group that were separated into more detailed classifications of resistance mechanism, these factors had varying degrees of contribution. For mutants resistant to ciprofloxacin (Supplementary Fig. 6a) and trimethoprim (Supplementary Fig. 6j), resistance mechanism was non-significant, but it was a significant factor for those resistant to mecillinam (Supplementary Fig. 6d) and nitrofurantoin (Supplementary Fig. 6g). Fitness was a significant factor only for the mecillinam resistance group (Supplementary Fig. 6e) and similarly, models containing both resistance mechanism and fitness were non-significant for all resistance groups, with the exception of the mecillinam-resistant mutants (Supplementary Fig. 6f).

In the first-step (GyrA) ciprofloxacin mutants, strain background was a significant factor for collateral responses

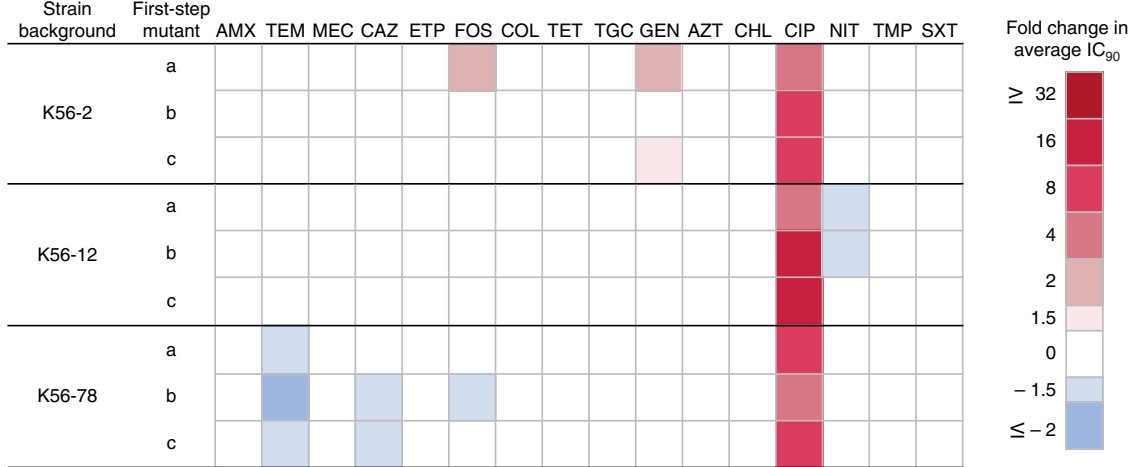

**Fig. 2** Collateral effects in *gyrA* mutants with decreased susceptibility to ciprofloxacin. Relative changes in antimicrobial susceptibilities, CS (blue) and CR (red), were determined by comparing average $IC_{90}$ values of nine first-step mutants to their respective wild-type strain. Antimicrobials are ordered by antimicrobial class, as in Supplementary Fig. 2

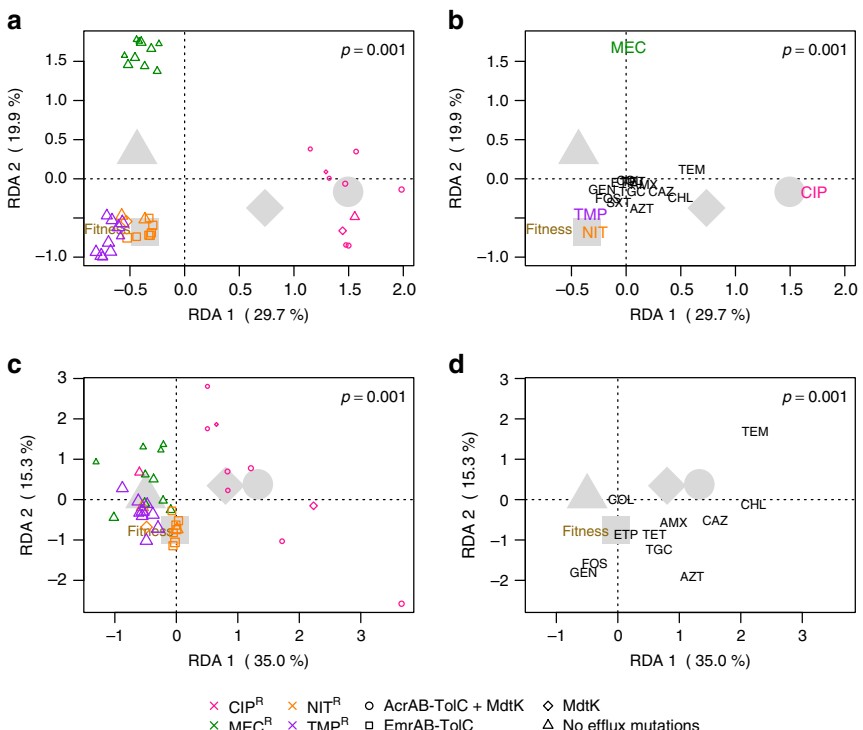

**Fig. 3** Results of multivariate statistical modeling. Graphical representations of two redundancy analyses (RDA, triplot) results relating various parameters to the observed changes in $IC_{90}$ between resistant mutants and respective wild-type strains for (**a, b**) 16 antimicrobials tested and (**c, d**) a subset of these antimicrobials, excluding ciprofloxacin, mecillinam, nitrofurantoin, trimethoprim, and trimethoprim-sulfamethoxazole. Each RDA is broken down into two plots; (**a, c**) where weighted averages of resistant mutants are plotted as colored symbols (color indicates resistance group, shape the assigned efflux group, and symbol size proportional to relative fitness, see Supplementary Fig. 4). In (**b, d**) antimicrobial drug names indicate the tip of vectors that pass through the origin in the direction of increasing $IC_{90}$ fold change or CR (direction of steepest ascent). Vectors can be used to interpret the change in $IC_{90}$ for the antimicrobials shown. For both statistical models, the first and second RDA axes shown display the majority of explained variation in $IC_{90}$ changes. Large gray symbols show centroids (average effect) for all resistant mutants within a given efflux group (shape). The vector tip of relative fitness (brown) is also shown. **a** The majority of explained variation is driven by primary resistances, where ciprofloxacin (pink)-resistant and mecillinam (green)-resistant mutants cluster distinctly from the other resistance groups, which showed higher relative fitness. **b** Resistant mutants possessing MdtK mutations alone (diamond) or together with AcrAB-TolC mutations (circle) are likely to show CR to chloramphenicol, ceftazidime, temocillin, and azithromycin, but sensitivity to gentamicin, fosfomycin, and trimethoprim. Whereas those without efflux mutations (triangle) are more likely to display low-level CS or no change to most antimicrobials tested. The analysis of the subset RDA (**c, d**) shows patterns consistent with the full model, but with less clustering of mutants by resistance group (**c**). The combination of AcrAB-TolC and MdtK efflux mutations displayed the greatest fitness costs, while mutants lacking efflux-related mutations were the most fit (**d**). RDA significance was assessed by permutation tests (1000 permutations), where $p \leq 0.05$ was considered significant. For more comprehensive multivariate models see Supplementary Fig. 5–6

(Supplementary Fig. 6m). However, this was not the case when the original ciprofloxacin-resistant mutants from the same strain backgrounds were added to the analysis (Supplementary Fig. 6n), suggesting again that other factors are more important than strain background. Overall, in comparison to the ciprofloxacin-resistant mutants, collateral responses of first-step mutants were far less frequent and more closely resembled those of the GyrA/ParC mutation-containing K56-2 CIP[R] mutant. A final redundancy analysis was performed on all ciprofloxacin-resistant and first-step mutants (Supplementary Fig. 6o), and showed a significant effect of resistance mechanism, supporting that mechanism, efflux in particular, is a major driver of collateral responses.

**Collateral responses shift the mutation selection window.** The mutant selection window can be defined as the concentration space between the lowest antimicrobial concentration that selects for and enriches resistant mutants[43] and the concentration that prevents the emergence of first-step resistant mutants, the mutation prevention concentration (MPC)[44,45]. In theory, if drug concentrations remain above the MPC during treatment, antimicrobial resistance is less likely to evolve[44,45]. It was recently demonstrated in *E. coli* MG1655 that changes in MPC correlated with collateral responses in resistant mutants[21]. We determined the MPC for 17 strain:drug combinations that exemplified conserved collateral responses (Fig. 1b). The MPC for each resistant mutant and its respective wild-type were compared. In 12/17 (70.6%) the change in MPC was consistent with the sign of collateral responses as determined by $IC_{90}$. This demonstrates that even small CS/CR changes can affect the mutant selection window, correspondingly shifting it down or up. In 4/17 (23.5%) the MPC displayed no change between the wild-type and mutant. This was observed when testing the MPC for mecillinam, trimethoprim, and azithromycin, though we speculate that increasing the precision of the MPC assay (as was done with $IC_{90}$ testing) might negate these discrepancies. Changes in MPC results with azithromycin were inconsistent with the change in $IC_{90}$ for a ciprofloxacin-resistant mutant and the mutants resistant to mecillinam and nitrofurantoin, which displayed a decreased MPC instead of an expected increase or no change, respectively.

## Discussion

Here, we identify conserved collateral responses in antimicrobial susceptibility across genetically diverse clinical *E. coli* strains following antimicrobial resistance development. Our findings are relevant beyond urinary-tract infections because uropathogenic *E. coli* are shown to also stably colonize the bladder and gut[46] and to cause bloodstream infections[47]. Our data show that CS and CR are pervasive in clinical *E. coli* strains, consistent with earlier results based on laboratory-adapted strains of various species[21–23,25,30,48] and a limited number of clinical isolates[21,30]. Resistance to ciprofloxacin resulted in a greater number of collateral responses than resistance to mecillinam, nitrofurantoin, or trimethoprim. This is likely due to mutations to known regulators of the AcrAB-TolC and MdtK efflux pumps. Both have broad substrate specificities to diverse antimicrobials including fluoroquinolones, β-lactams, tetracycline, chloramphenicol, trimethoprim-sulfamethoxazole, and some macrolides for the AcrAB-TolC efflux pump[49,50], and fluoroquinolones, chloramphenicol, trimethoprim, and some β-lactams for the MdtK pump[39,51]. Interestingly, both overexpression of MdtK[51] and RpoB[39] mutations (that were linked to MdtK expression) have been shown to reduce susceptibility to fosfomycin, as was observed in the ciprofloxacin-resistant mutants in this study (Fig. 1a). Overall, CR was much more prevalent than CS, and the magnitude of

collateral responses were most often small, consistent with other reports[21–23]. We observed that collateral responses varied substantially by resistance group, but variation was also observed within resistance groups.

Using $CS_{50}$ and $CR_{50}$ thresholds to identify conserved responses, we found that conserved CR was more than twice as common as conserved CS. Whereas many of the conserved collateral responses identified in this study support the findings in previous work using single laboratory-adapted strains, we observed several clinically relevant differences. For example, our finding of conserved CS in ciprofloxacin-resistant mutants to gentamicin was previously reported in *E. coli* K12[22] but not in *E. coli* MG1655[21]. In mutants resistant to ciprofloxacin we also observed conserved CR towards chloramphenicol, as reported in ref. [21], but not in ref. [23]. We identified conserved CR of nitrofurantoin-resistant mutants to amoxicillin, and this was not reported in MG1655[21]. These observations underscore the importance of exploring collateral networks in multiple mutants of different clinical strain backgrounds and with different resistance mechanisms to assess their potential clinical application.

Visual inspection of the data revealed a few clinically relevant examples of CS phenotypes that appeared independent of putative mechanism of resistance. We show that *E. coli* strains resistant to ciprofloxacin display CS towards gentamicin, fosfomycin, ertapenem, and colistin, and these phenotypes were conserved across multiple mechanisms of resistance. These results parallel those of a recent study on *P. aeruginosa* clinical isolates from cystic fibrosis patients, where resistance to ciprofloxacin was associated with CS to gentamicin, fosfomycin, and colistin[31]. Taken together these data support the presence of general, conserved collateral networks that may both affect the population dynamics of antimicrobial resistance during treatment and counter-select for resistance, as recently indicated[31].

We assumed a priori that genetic background, resistance group, resistance mechanism, and the fitness cost of resistance could potentially affect the generality, sign, and magnitude of collateral networks in clinical *E. coli* strains. Despite the fact that some collateral responses are conserved across different strains and mechanisms of resistance, our multivariate statistical approaches show overall that mechanism of resistance is the key predictor of CS and CR variability. This is primarily the case for efflux-related mutations. However, mechanism of resistance also significantly contributed to the observed CS and CR variation in the mecillinam mutants where no efflux mutations were found. The presented data are consistent with earlier reports based on multiple resistant mutants derived from single strains with different resistance mechanisms towards specific antimicrobials[22,23,52]. Our finding that genetic background did not significantly contribute to collateral responses is an important addition to these earlier studies. Finally, we found that the fitness cost of resistance also contributed significantly to the observed variation in CS and CR, despite some collinearity between efflux-related mutations and reduced fitness. Taken together, our data and previous reports indicate that applied use of collateral networks in future treatment strategies may be dependent on rapid identification of specific resistance mechanisms. Moreover, clinical application of CS as a selection inversion strategy warrants further investigations to ideally explore CS in isogenic backgrounds, representing several diverse strains, with permutations of all known antimicrobial resistance-associated traits. Such extensive studies would likely provide valuable information on the mechanisms of CS. Other confounding factors such as mobile genetic elements with heterogeneous resistance determinants should also be investigated as they would likely influence and reduce the predictability of collateral networks.

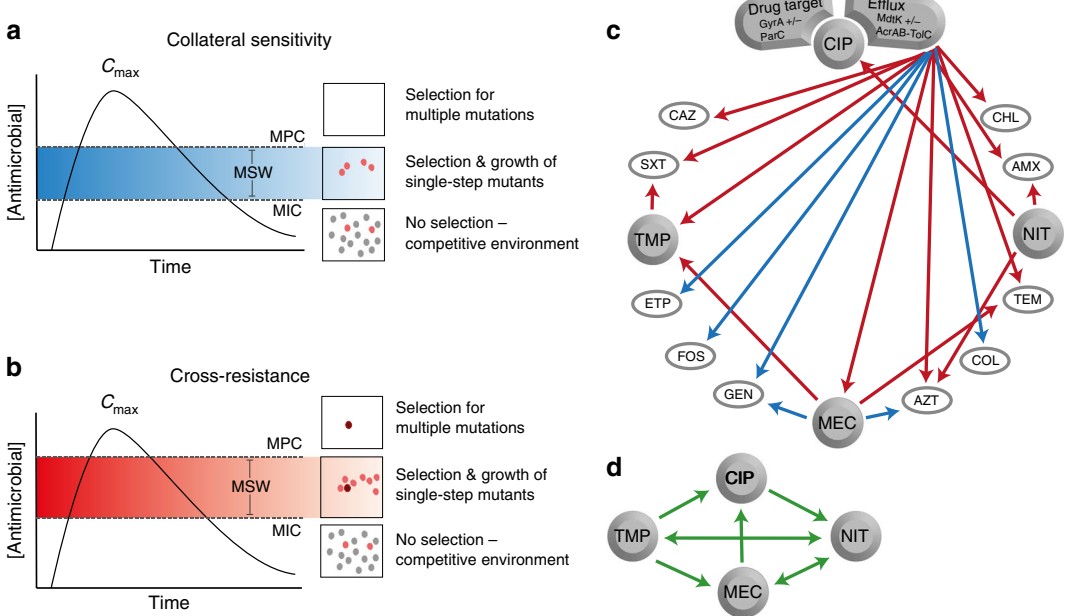

**Fig. 4** Presentations of the potential effects and implications of CS and CR. **a** Sequential drug administration informed by CS (blue) could potentially narrow or shift the mutant selection window (MSW) downwards in concentration space whereas (**b**) CR (red) results in a widened or shifted upwards mutant selection window for secondary antimicrobials. This would affect the probability of acquiring second-step mutations leading to high-level resistance. Consequently, CS-informed secondary therapies could reduce selection and thus propagation of first-step mutants resulting in a reduced opportunity for second-step mutations to occur. Dots represent bacteria resistant to a primary antibiotic (gray), spontaneous mutants with reduced susceptibility to a secondary drug (pink), or those with high-level resistance to the secondary drug (dark red). Note that these are hypothetical schematics and in many cases the maximum concentration achieved ($C_{max}$) may be below the MPC. **c** Arrows indicate conserved collateral responses, where CS (blue) and CR (red) are depicted. The collateral responses in this study are mainly predicted by efflux-related mutations in the ciprofloxacin-resistant mutants. These data suggest potential secondary treatment options that may reduce the rate of resistance evolution (**a**, **b**) following initial treatment failure. **d** Green arrows indicate putative temporal administration of four antimicrobials used for the treatment of urinary-tract infections, as informed by the collateral networks in (**c**)

Selection inversion, as described by ref. [21], depends on the cycling of drug pairs that display reciprocal CS. We did not observe reciprocal CS between any of the four drugs studied here that are widely used for treatment of urinary tract infections. However, we asked if modest reductions and increases in antimicrobial susceptibilities would affect the mutant selection window[44] for the most prevalent CS and CR phenotypes. We subjected conserved CS and CR phenotypes to MPC assays and revealed that even a small 1.5-fold change in $IC_{90}$ could equally alter the MPC, resulting in a shift of the mutant selection window (Fig. 1b). These results suggest that antimicrobial treatment strategies informed by collateral networks could affect the evolutionary trajectories of antimicrobial resistance. Sequential treatment using drug pairs that display CR would, following resistance development, shift the mutant selection window towards higher antimicrobial concentrations, as was previously observed[53], and increase the likelihood for resistance development to subsequent treatment options (Fig. 4a). Conversely, sequential treatment based on drug pairs that display CS can shift the mutant selection window down and reduce the window of opportunity for high-level resistance development (Fig. 4b). This result suggests that the initial choice of antimicrobial may set the stage for later resistance development.

Based on our in vitro findings, trimethoprim and nitrofurantoin are attractive from a clinical perspective, as resistance to these resulted in few collateral responses, preserving the innate sensitivity to available secondary antimicrobials (Fig. 4c, d). However, mecillinam could be even more attractive, as CS largely dominates the observed collateral responses in resistant mutants. Additionally, isolates resistant to mecillinam,

especially those evolved in vivo, are associated with high fitness costs[42]. In contrast, exposure to ciprofloxacin was more likely to cause dramatic collateral responses that depend on the mechanism of resistance and could potentially negatively impact future therapeutic options (Fig. 4c). These observations align with antimicrobial treatment recommendations in Norway, where mecillinam, nitrofurantoin, and trimethoprim are recommended for first-line therapy of uncomplicated urinary-tract infections, and ciprofloxacin is reserved for otherwise complicated infections[54]. Similarly, in the United States nitrofurantoin, trimethoprim-sulfamethoxazole, and mecillinam are recommended before fluoroquinolones, such as ciprofloxacin, ofloxacin, and levofloxacin[55].

Our conclusions are not without limitations. First, we acknowledge that including more clinical isolates from different infection foci, more diverse genetic backgrounds including different bacterial species, as well as other selective agents, could change the outcome of our statistical analyses. This would allow increased sensitivity for the assessment of the different factors controlling collateral responses. A more targeted approach to assess the impact of specific resistance mechanisms on CS and CR across genetically diverse clinical strains is lacking in the field. Our analyses suggest that the fitness cost of resistance explains some variability in the collateral networks reported here. We used relative growth rates as a proxy for relative fitness, and our data are consistent with reports demonstrating that growth rates affect susceptibilities to several antimicrobials[56,57]. It is unclear if collateral networks will be perturbed by compensatory evolution, which eliminates the fitness costs of primary resistance[58–60]. Finally, this and previous studies focus on antimicrobial

resistance development to a single drug, and there is a complete lack of data on how multidrug resistance, including resistance genes on mobile genetic elements, will affect collateral networks. We are currently investigating these and other questions that will aid in our understanding of collateral networks and their potential therapeutic application.

## Methods

**Bacterial strains**. We used 10 clinical, urinary-tract infection isolates of *E. coli* from the ECO-SENS collections[61,62] originating from countries across Europe between 2000 and 2008 (Table 1). The isolates were chosen to represent pan-susceptible strains with diverse genetic backgrounds and were reported plasmid-free[32]. Subsequent analysis based on whole-genome sequencing discovered two changes to previously reported sequence types and the presence of plasmid replicons in three strains (Supplementary Table 4). *E. coli* ATCC 25922 was used for reference and quality control purposes. For general growth, bacterial strains were grown in either Miller Difco Luria–Bertani (LB) broth (Becton, Dickinson and Co., Sparks, MD, USA) or on LB agar; LB broth with select agar (Sigma-Aldrich) at 15 g L$^{-1}$ and incubated at 37 °C.

**Selection of antimicrobial-resistant mutants**. Single antimicrobial-resistant mutants were selected at drug concentrations above the European Committee on Antimicrobial Susceptibility Testing (EUCAST) clinical breakpoints[63] for ciprofloxacin, nitrofurantoin, trimethoprim, and mecillinam (Supplementary Table 1). Briefly, 100 μL of 10× concentrated overnight culture was spread on Mueller Hinton II agar (MHA-SA; Sigma-Aldrich) plates containing ciprofloxacin (Bio-chemika), nitrofurantoin (Sigma-Aldrich), or trimethoprim (Sigma-Aldrich) with two-fold increasing concentrations of the antimicrobial. After 24–48 h, a mixture of growth from the highest concentration plate with multiple colonies was used to start a new overnight culture at the same antimicrobial concentration. This was repeated until there was growth above the clinical breakpoint[63]. Mecillinam-resistant mutants and first-step ciprofloxacin mutants (*gyrA* mutation-containing) were selected as single-step mutants on LB or MHA-SA agar, respectively. Mutants were confirmed as *E. coli* using matrix-assisted laser desorption ionization time-of-flight (MALDI-TOF) analysis with MALDI BioTyper software (Bruker, MA, USA).

**Antimicrobial susceptibility testing**. Mutants were initially screened for resistance above EUCAST breakpoints[63] with gradient diffusion strips following manufacturers guidelines (Liofilchem, Italy), on Mueller Hinton II agar (MHA-BD; Becton, Dickinson and Company). Plates with insufficient growth were incubated for an additional 24 h.

To maximize the precision of our susceptibility estimates and in accord with related studies on CS, collateral changes were determined by IC$_{90}$ testing[21] with some modifications. IC$_{90}$ values were determined following 18 h incubation at 700 rpm (3 mm stroke) in Mueller Hinton Broth (MHB, Becton, Dickinson and Co.). Slow-growing strains where positive growth controls did not reach OD$_{600\,nm}$ of 0.3 after 18 h (i.e. K56-12 CIP$^R$) were interpreted after 42 h incubation. Standard two-fold concentrations and the median values between them were used as a 1.5-fold testing scale. IC$_{90}$ values were the lowest concentration tested that resulted in ≥90% inhibition of growth. Percent inhibition was calculated compared to the positive control (untreated sample) with background removed[21]. IC$_{90}$ results were determined in at least three biological replicates on separate days. The final result reflects the average of a minimum of three replicates that met quality control standards, including the result of ATCC 25922 on each plate, positive growth control OD$_{600\,nm}$ > 0.3, negative growth control OD$_{600\,nm}$ < 0.05, and accepting no more than one skip (defined as a break in the inhibition pattern). When one skip was observed, the IC$_{90}$ value was consistently interpreted as the lowest concentration tested that resulted in ≥90% inhibition of growth. Fold change in IC$_{90}$ was calculated as the ratio between the resistant mutant and its respective wild-type. The IC$_{90}$ testing varied for two antimicrobials (according to EUCAST recommendations), where fosfomycin was tested in MHB supplemented with 25 μg mL$^{-1}$ glucose-6-phosphate (Sigma-Aldrich) and tigecycline was tested in fresh MHB media that was prepared daily.

Dose–response curves were generated with average OD$_{600}$ values (background subtracted) for concentrations tested during IC$_{90}$ testing. Averages were plotted for mutants and respective wild-type strains.

**MPC testing**. We determined the MPC for 17 of 20 conserved collateral responses. Temocillin was excluded due to lack of supply and trimethoprim-sulfamethoxazole was excluded for trimethoprim-resistant isolates due to fundamental CR between trimethoprim and trimethoprim-sulfamethoxazole. MPC determination was based on previous work by Marcusson et al.[64]. Briefly, 10 mL overnight culture was centrifuged and the pellet re-suspended in 1 mL MHB, estimated to contain ≥10$^{10}$ CFU (actual values were $1.4 \times 10^{10}$–$7 \times 10^{10}$ CFU). The inoculum was split and spread onto four large (14 cm diameter) MHA-SA agar plates for each antimicrobial concentration tested in a two-fold dilution series. The MPC was the lowest concentration with no visible growth after 48 h. Where growth/no growth was difficult to interpret, suspected growth was re-streaked on plates at the same

antimicrobial concentration. Azithromycin and ertapenem were regularly inconsistent, making re-streaking essential. Resistant mutants and wild-types were tested in parallel, and results represent the average of at least two biological replicates.

**Growth rate measurements**. To obtain growth curves of wild-types and resistant mutants, single colonies were used to inoculate at least three biological replicates of MHB starter cultures (2 mL) that were incubated at 37 °C for 24 h shaking at 500 rpm. Each culture was diluted 1:100 in MHB (resulting in ~$2 \times 10^7$ cell mL$^{-1}$) and 250 μL was added in triplicate to a 96-well microtiter plate. The plate was incubated overnight at 37 °C in a Versamax plate reader (Molecular Devices Corporation, California, USA) with shaking for 9.2 min between reads. OD$_{600}$ measurements were taken every 10 min and growth rates were estimated using the GrowthRates v.2.1 software[65]. To a varying extent, the ciprofloxacin-resistant mutants of K56-12, K56-16, K56-44, and K56-68, as well as, K56-44 MEC$^R$, K56-68 MEC$^R$, and K56-70 TMP$^R$ displayed noise in the growth curves due to clumping or non-homogeneous growth, and GrowthRates was unable to fit a line with *R*-value above the 0.98 cut-off value. Additional experiments and visual inspection of the log-transformed OD$_{600}$ values were used to solve this issue. If GrowthRates failed to analyze the curves, a line was fitted within the log phase through at least five consecutive points that displayed log-linear growth. Growth rate (*r*) was calculated based on the slope[65]. Relative growth rates were calculated as $r_{(resistant\ mutant)} \, r_{(wild-type)}^{-1}$.

**Whole genome sequencing**. Genomic DNA was isolated using the GenElute Bacterial Genomic DNA kit (Sigma-Aldrich) following guidelines for Gram-positive DNA extraction. Purity and quantification was determined with Nanodrop (Thermo Scientific) and Qubit High Sensitivity DNA assay (Life Technologies), respectively. For library preparation of wild-types and resistant mutants, 1 μg of DNA was sheared on a Covaris S2 to ≈400 bp using the recommended settings (intensity: 4, duty cycle: 10%, cycles per burst: 200, treatment time: 55 s). Libraries were then prepared and indexed using the DNA Ultra II Library Preparation Kit (New England Biolabs, E7645). For library preparation of first-step ciprofloxacin mutants, 1 ng of DNA was used with the Nextera XT DNA prep kit (Illumina, San Diego), according to the producer's instructions. All libraries were quantified by Qubit High Sensitivity DNA assay and distributions and quality assessed by Bioanalyser DNA 1000 Chip (Agilent, 5067-1504) before normalizing and pooling. Illumina sequencing, NextSeq 550, and MiSeq, was used for the first-step ciprofloxacin mutants and the wild-types and resistant mutants, respectively, using paired-end reads. For NextSeq 550 a mid-output flowcell with 300 cycles was used. V2 chemistry was used for the MiSeq sequencing.

Wild-type genomes were assembled, as follows. Illumina adapters on wild-type reads were removed with Trimmomatic version 0.36[66] using standard settings, then assembled with SPAdes[67]. Contigs less than 500 bp in length or less than 2.0 coverage were removed. Wild-type genome assemblies were inspected and compared to the *E. coli* MG1655 genome (GenBank U00096.2) using QUAST[68]. Final assembled genomes of wild-type strains were annotated using the automated Prokaryotic Genome Annotation Pipeline (https://www.ncbi.nlm.nih.gov/genome/annotation_prok/).

Wild-type and mutant sequences were compared to identify putative resistance mechanisms. First, wild-type genomes were annotated with Rapid Annotation using Subsystem Technology server (RAST, version 2.0) for *E. coli*[69]. SeqMan NGen (DNASTAR, Madison, WI) was used to align raw mutant reads to the corresponding, annotated wild-type genomes, using standard settings. Reported SNPs had ≥10× coverage depth and ≥90% variant base calls. SNPs present in the wild-type assembly or in at least two mutants resistant to different antimicrobials from the same strain background were excluded. Reported SNPs, indels, and rearrangements were manually inspected and gene annotations confirmed using Gene Construction Kit (Textco Biosoftware Inc., Raleigh, NC) and NCBI BLAST searches, respectively.

Multilocus sequence typing (MLST), as well as, plasmid replicon and acquired antimicrobial resistance gene content were determined for the wild-type genomes using MLST version 1.8, PlasmidFinder version 1.3, and ResFinder version 3.0[70], respectively (http://genomicepidemiology.org/). The MLST of each wild-type strain was confirmed compared to the original reported sequence type[32] for all but two strains, K56-41 and K56-70. These strains were originally described as ST420 and ST550, but were ST73 and ST537 in our analysis, respectively (Supplementary Table 4). The wild-type strains were previously described as plasmid free, but we identified two small plasmid replicons, Col156 in K56-16 and K56-44 and Col (MP18) in K56-75 (Supplementary Table 4). Using ResFinder, we detected only one acquired genetic element, *sul*2 (linked to sulfonamide resistance) in K56-44, and two point mutations, PmrB V161G in K56-50 and K56-70 and ParE D475E in K56-78, that are linked to colistin and quinolone (ciprofloxacin) resistance, respectively (Supplementary Table 4). Though all of the wild-type strains were phenotypically pan-susceptible (Supplementary Fig. 1c), these resistance determinants could affect antimicrobial susceptibilities differentially in the presence of other mutations[71,72].

To assess genetic diversity, a phylogenic tree was generated based on the genome sequences of wild-type strains. Assembled wild-type genomes were annotated with PROKKA[73], the core gene-encoding regions were extracted and compared using ROARY[74], and a maximum-likelihood tree with 100 bootstraps

was generated using RAxML[75]. Genetic distances were calculated in R[76] using previously described methods[77].

**Multivariate statistical analyses.** The fold changes of mean $IC_{90}$ values relative to the parental wild-type strain (collateral responses) were log transformed. Statistical analyses were performed on the complete data set, as well as a subset of the data excluding five antimicrobials (ciprofloxacin, mecillinam, nitrofurantoin, trimethoprim, and trimethoprim-sulfamethoxazole). To estimate and test the effects of strain background, resistance group, resistance mechanism, growth rate, and relative fitness we relied on multivariate modeling (redundancy analysis) to address the co-variation in $IC_{90}$ across antimicrobials. A redundancy analysis is a constrained version of a principle component analysis that additionally allows for hypothesis testing. Linear constraint scores were plotted for each mutant. Response variables were overlaid with independent scaling to illustrate the direction of steepest ascent (increasing CR) from the origin for each antimicrobial. Data were inspected to check whether the assumptions underlying redundancy analysis were met. Significance testing of multivariate models and their factors (Supplementary Table 3) was done via permutation tests (1000 permutations), an approach robust to deviations from multivariate normality and variance homogeneity; $p < 0.05$ was considered significant. Analyses were done in R[76] using the Vegan work package[78].

## Data availability

Whole-genome sequencing data are available at NCBI (BioProject PRJNA419689). All other relevant data are available within this article, the Supplementary Information, or from the corresponding author upon request.

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

## Acknowledgements

We thank Tony Brooks at UCL Genomics and Hagar Taman at UiT Genomics Support Centre Tromsø for genome sequencing, Ane L.G. Utnes for generation of nitrofurantoin-resistant mutants, Jessica N. Tran for generation of first-step ciprofloxacin mutants, Jessin Janice for phylogenetic analysis, Søren Overballe-Petersen for pre-liminary processing of wild-type genomes, and Maria Chiara Di Luca for analysis of the genomes for MLST and identification of plasmid replicons. Funding for this project was provided through the Northern Norway Regional Health Authority and UiT—The Arctic University of Norway (Project SFP1292-16), and JPI-EC-AMR (Project 271176/H10).

## Author contributions

P.J.J. and Ø.S. conceived the project; N.L.P., E.G.A.F., J.K. and V.S. designed and performed experiments; R.P. designed and R.P. and N.L.P. performed the multivariate statistical modeling; all authors analyzed, interpreted and discussed the data; and N.L.P., E.G.A.F., A.P.R., D.E.R. and P.J.J. wrote the manuscript with contributions from the other authors.

## Additional information

**Competing interests:** The authors declare no competing interests.

