## [Peer Review File · Nature Communications]

Reviewers' comments:

Reviewer #1 (Remarks to the Author):

MAIN COMMENTS

A. The main point of this paper is that resistance to one antimicrobial can increase or decrease the propensity for resistance to emerge to other antimicrobials. The approach developed can guide antimicrobial selection and order of treatment to minimize the emergence of resistance. That could be important. The paper would be stronger and more citable if specific clinical advice is derived. Reaching this goal may require stronger experiments, an issue the authors can address.

B. Use of IC-90 may not be optimal, as IC-50 may be a more accurate parameter. Please consider probit analysis, change to IC-50 for analysis. If I am correct, this will increase the rigor of the work and author credibility.

C. Ciprofloxacin appears to be an especially problematic compound. You will need to supply a good explanation, which probably involves selection of efflux mutations. You may not be able to say why quinolones are prone to efflux, but documenting that they are special will be satisfying. Compare with other antimicrobial classes.

D. Efflux to quinolones is likely selected at low doses. There are several examples in which population analysis shows that low doses select so many non-target mutants that target mutant are not seen (see Zhou 2000 J Inf Dis for mycobacteria). While these non-target mutants are likely to be efflux, but that was probably not shown. There should be other papers that will make this point about efflux. Since many of the cip-resist mutants in the present study were obtained by multiple rounds of challenge, it is not surprising that many efflux mutations will be present. As a test, select single-step *gyrA* mutants by very high concentrations on ciprofloxacin. They should not have efflux mutants and they should behave differently with respect to resistance to other antimicrobials. This is an important experiment because it supports the general idea that dosing high is superior to dosing low and using marginally active fluoroquinolones. Thus, this experiment will add an important dimension to the paper.

E. Effects on the mutant selection window are mentioned and documented in a figure. This is an important support for the selection window hypothesis, and that could be mentioned. Also note that the accumulation of resistance alleles raises the mutant selection window has been reported and cited (Li 2004 AAC 48: 4460). There may be other examples.

F. Although English usage is excellent, the excessive use of acronyms slows reading. At every acronym, other than common ones such as DNA, reading stops as the reader tries to remember the definition. The only reason for using the acronyms is to keep the word count down, but at the cost of reader understanding. The work could also be made more readable by shorter paragraphs. The material is complex, and the reader needs spots to consolidate the information. More "Thus" sentences would help the flow of the story. Comma errors and hyphenation errors with compound adjectives suggest that the work could have been more carefully prepared.

SPECIFIC, OFTEN MINOR COMMENTS (some may be redundant with the above; items with asterisk are important to consider)

Line 50: I would like to see a reference for the statement that the rate is lagging.

Line 105: should you mention that the single mutants were, in many cases, obtained by repeated challenge to clarify the method?

*Line 114: please explain why you used IC-90 rather than IC-50. I would expect plots of IC-50, due to the shape of the curve, to be more sensitive to small differences than IC-90. There is a large, old literature dealing with this issue and use of all data points to estimate IC-50. Since you probably have the data, you may want to reconsider your assay and perhaps consider probit analysis. While this issue may not invalidate the overall conclusions, it seems that you could be more rigorous, which would improve author credibility.

Line 121 networks. Are you sure this is the right/best word? For most readers networks will mean sets of interacting genes that are identified by the work. You may be misleading the reader here.

Line 132 changes does not mean only reduced – changes could mean increased.

Line 156 clinically relevant. This term is vague, since the changes could influence the emergence of resistance (rather than cure rate), which would be clinically relevant.

Line 158. Do you offer an explanation for the unusual behavior of ciprofloxacin-resistant mutants? I would minimize thinking on the part of the reader.

Line 164 compound adjective requires hyphen

Line 166 resistance. I think you mean susceptibility, since you are not dealing with breakpoints here (breakpoints are key to the definition of resistance)

Line 190. These data seem to be largely expected from previous work. Do you want to say “As expected”? I think it is good to distinguish what is really new in the present work from what is basically confirmatory.

Line 201 pan-susceptible by growth criteria. But you are looking at the emergence of additional alleles, and there are reports in the literature that par mutations can dramatically increase the selection of quinolone resistant *gyrA* alleles (see Zhao PNAS 1998). Thus, you need to rethink your statement here.

Line 225: limited is vague. Do you mean minor? Non-significant – do you mean insignificant? Or do you mean not significant with respect to a statistical parameter?

Line 226. Is your data really accurate to three significant figures? Also, the meaning of this number requires comparison with results from mutations (reiterate if stated above)

Line 254 and elsewhere you need to define significant, perhaps with p values.

Line 281. How did you demonstrate that the isolates were genetically diverse? What is your definition of genetically diverse? One line 387 you fail to specify. Is diversity expected because the isolates were from different countries? That seems like a weak argument. This is not a trivial point, since even with whole genome sequencing you have a spectrum. Thus, you will need some definition. Or rephrase genetic diversity to lack definition and still try to maintain credibility.

Line 375: unknown. Since you use the word action, you are covering both bacteriostatic and bactericidal effects. The word unknown is misleading with respect to lethal action, largely because growth rate, in particular stationary phase, can reduce the production of reactive oxygen species which are now well-established as contributors to antimicrobial-mediated killing. A recent reference on growth phase is Collins Mol Cell for mechanism for a phenomenon that dates back at least to the 1960s.

Table S1. Single replicate. If the data used in the analysis derived from multiple replicates, mention that. Also, spelling of drugs in footnotes.

*Table S4. DNA mutations. Is there another kind? I cannot believe that your data are significant in some cases to 6 significant figures. These numbers suggest a lack of understanding of significant figures and undermines author credibility.

Line 109 You might want to mention that in some cases multiple rounds of selection were used.

*Fig 4. The situation portrayed here, drug concentrations in patients exceeding MPC, has been reported for only a few drug-pathogen combinations (one is ciprofloxacin vs *H. influenzae*; another is moxifloxacin vs *S. pneumoniae*). In general, C_{max} is inside the selection window. The authors should be careful, perhaps by redrawing the figure or by modifying the legend, to avoid misleading the reader.

Reviewer #2 (Remarks to the Author):

The manuscript by Podnecky et al describes the collateral sensitivity networks of the *E. coli* strains from UTI. Single isolates for each drug resistant populations were tested against the panel of 16 antibiotics to determine the change in susceptibility. Resistance mechanisms and mutant selection windows were further investigated as well.

Major comments

The study expands the number of *E. coli* isolates tested from 3 strains (Imamovic and Sommer, 2013) to 10 clinical strains in the current study. While the research performed by Podnecky and colleagues expands to broader collection of isolates than used before for *E. coli*, the novelty of this study is somewhat limited. The major driver for the study was said to be not explored direction for genetically diverse strains. However, the authors only use 10 strains as well, this small collection of isolates it is only limited to pan-susceptible strains as well. The authors could offer further information for strain selection. How diverse these strains are (e.g. number of SNPs)?

In particular the term used "in depth study" across genetically diverse isolates seems over statement. All isolates were susceptible to antibiotics and such scenario does lead to the misleading conclusion no matter what the genetic background is, it would lead to conserved responses (as also recognised by authors in the discussion section).

Minor comments

Line 299. *E. coli* MG1655 is the *E. coli* strain as well. Please specify the strain differences correctly from the appropriate references quoted, if any.

Reviewer #3 (Remarks to the Author):

The topic of the paper is very timely and important; collateral-sensitivity (CS) informed therapy is a promising selection inversion strategy to combat antimicrobial resistance. With this work the authors intend to fill the research gap regarding the conservation of CS networks across diverse clinical *E. coli* strains. To this purpose i) they chose ten different UTI clinical *E. coli* isolates, ii) generated antimicrobial resistant mutants from each of these isolates with resistance levels above clinical breakpoints against four different antibiotics used in UTI treatment and iii) they investigated the collateral networks of the resistant mutants against 16 antimicrobials from different drug classes.

By performing a multivariate statistical analysis they found that genetic background did not contribute significantly to collateral responses and based on this they claim that collateral responses are conserved across different *E. coli* clinical strains. In addition to this their main conclusion is that resistance mechanisms, especially efflux related mutations are the key predictors of collateral responses.

Major concerns:

Conservation of the collateral responses across diverse clinical E. coli strains is the main result of this paper. To be able to draw this important conclusion the authors need to present a figure with the phylogenetic tree of E. coli strains where they place the ten studied E. coli isolates and show that indeed these are different from each other. This is indispensable to draw a conclusion about conservation.

Please note also that the manuscript investigates E.coli strains only. It is unknown whether the same results would hold for other enterobacterial species, such as Salmonella.

Their second main conclusion is that efflux mutations are the key predictors of collateral responses. However, when they performed the multivariate analysis separately for each AMR group this conclusion was not valid for trimethoprim and ciprofloxacin resistant mutants. The authors should give a possible explanation for this, especially because almost half of the conserved collateral responses were found in ciprofloxacin resistant mutants.

At the end of the abstract the authors claim that “detection of resistance mechanisms is important to accurately predict collateral antimicrobial responses”, but they do not present a “prediction map” based on their results. Put it simply, they should give some examples that the presence of given mutations in resistant strains would indicate collateral sensitivity or collateral resistance against given antibiotics.

Minor concerns:

- Elaborate on the reason of choosing redundancy analysis (RDA) for multivariate statistical analysis.
- In my opinion figure 2 contains too much information making the interpretation difficult. It would be helpful for the reader if the authors could dissect it into separate figures, each highlighting one main conclusion.

In sum, if the authors address the raised concerns in a satisfactory way, I support the publication of the paper in Nature Communications.

Response to reviewers:

Reviewer #1 (Remarks to the Author):

MAIN COMMENTS

A. The main point of this paper is that resistance to one antimicrobial can increase or decrease the propensity for resistance to emerge to other antimicrobials. The approach developed can guide antimicrobial selection and order of treatment to minimize the emergence of resistance. That could be important. The paper would be stronger and more citable if specific clinical advice is derived. Reaching this goal may require stronger experiments, an issue the authors can address.

Our response: We jointly address this and a similar comment made by reviewer 3 on the inclusion of a prediction map informed by mechanisms of antimicrobial resistance. In the revised version of this manuscript we include an updated version of Figure 4 that proposes how conserved CS and CR responses reported here for E. coli can be used to inform options for secondary treatment. These predictions are strengthened in this revised version of the manuscript by the addition of experimental data on single step gyrA mutants and how they affect collateral responses (see below). As we consider the concept of CS/CR to require appropriate clinical evaluation, we are reluctant to give specific clinical advice and rather include pre-clinical recommendations without trial data. See Figure 4c-d and discussion on lines 410-418.

B. Use of IC-90 may not be optimal, as IC-50 may be a more accurate parameter. Please consider probit analysis, change to IC-50 for analysis. If I am correct, this will increase the rigor of the work and author credibility.

Our response: We have carefully considered this point and respectfully disagree. We have decided to leave the IC-90 analyses, as is in the manuscript, for two main reasons:

- 1. Collateral effects reported in the existing literature are quantified using different approaches, but most often these report IC90 or MIC. To enable direct comparison with this work, we chose IC90 because it reflects the majority of recent CS papers (see references below) but also because this method is most easily translated to the setting of clinical labs using similar approaches via standard MIC determination. Please see:***

- 1) Imamovic and Sommer. 2013 Sci Transl Med 5: 204ra132***
- 2) Munch et al. 2014 Sci Transl Med 6:262rara156***

- 3) *Rodriguez de Egrafov et al. 2015. Mol Biol Evol 32:1175-85*
- 4) *Linkevicius, Sandegren, and Andersson 2016. Antimicrob Agents Chemother 60: 789-86*
- 5) *Imamovic et al. 2018 Cell. 172: 121-34*

2. Importantly, using *E. coli*, Munch and co-workers (reference 2 above) found that there is a significant linear relationship between IC50 and IC90 (as well as for MIC and IC90), see Fig S3 in the PMC free version at <https://www.ncbi.nlm.nih.gov/pmc/articles/PMC4503331/>. These data make clear that the conclusions of our work are not affected by our choice to use IC90.

C. Ciprofloxacin appears to be an especially problematic compound. You will need to supply a good explanation, which probably involves selection of efflux mutations. You may not be able to say why quinolones are prone to efflux, but documenting that they are special will be satisfying. Compare with other antimicrobial classes.

Our response: We agree that it is worrying that exposure to ciprofloxacin can lead to such a wide repertoire of CR to many classes of clinically important antimicrobials. In the revised manuscript, we now discuss the well-documented evidence that the AcrAB-TolC efflux pump has broad substrate specificity. In addition to ciprofloxacin it is known to efflux other fluoroquinolones, beta-lactams, tetracyclines, trimethoprim-sulfamethoxazole and some macrolides (Poole, K. 2004 Clin Microbiol Infect; Elkins, CA, Mullis, LB. 2007 Antimicrob Agents Chemother). Similarly, MdtK-mediated efflux affects fluoroquinolones, chloramphenicol, trimethoprim, some beta-lactams (Nishino, K, Yamaguchi, A. 2001 J Bacteriol; Pietsch, F, Bergman JM, et al. 2016 J Antimicrob Chemother), however since this is a newly described mechanism of ciprofloxacin resistance it is less well characterized. Please see lines 328-336.

D. Efflux to quinolones is likely selected at low doses. There are several examples in which population analysis shows that low doses select so many non-target mutants that target mutant are not seen (see Zhou 2000 J Inf Dis for mycobacteria). While these non-target mutants are likely to be efflux, but that was probably not shown. There should be other papers that will make this point about efflux. Since many of the cip-resist mutants in the present study were obtained by multiple rounds of challenge, it is not surprising that many efflux mutations will be present. As a test, select single-step *gyrA* mutants by very high concentrations on ciprofloxacin. They should not have efflux mutants and they should behave differently with respect to resistance to other antimicrobials. This is an important experiment because it supports the general idea that dosing high is superior to dosing low and using marginally active fluoroquinolones. Thus, this experiment will add an important dimension to the paper.

Our response: We agree that an experiment investigating single-step gyrA mutants would add an important dimension to this manuscript. As recommended, we have therefore performed it. As reviewer 1 (and we) expected, single gyrA mutants uniformly did not display most of the CR responses that were seen in the multi-step ciprofloxacin resistant mutants. This strongly supports a link between the widespread occurrence of CR in the ciprofloxacin resistant mutants and the observed efflux mutations. This adds important mechanistic support for the idea that efflux dictates most collateral effects as now described on lines 207-216 and 285-295.

In response to the first point about efflux selected at low doses: The evolutionary trajectory of ciprofloxacin resistance in E. coli is typically initiated by gyrA target mutations yielding an MIC below the clinical breakpoint. After this initial step two trajectories can be taken. 1) gyrA plus additional non-target efflux mutations, and 2) gyrA plus parC (this is the most common genotype described in clinical strains). In the laboratory the emergence of the different mutations is highly conditional. We have observed that static selection at increasing doses predominantly leads to non-target efflux mutations and the frequently described gyrA and parC combination is rarely observed. The high rate of gyrA target mutations in combination with several different off-target efflux mutations reflects the much greater mutational target size as compared to parC mutations. Moreover, the gyrA and parC combination has a much lower fitness cost than compared to the gyrA and efflux combination. Consequently, selection for the most prevalent gyrA and parC combination requires weak prolonged (multi-generation) selection in order for these rare mutants to outcompete the more prevalent gyrA and efflux mutants. The evolutionary dynamics of ciprofloxacin resistance in E. coli was elegantly characterized and described by the Hughes-group in Uppsala (Huseby, DL, Pietsch, F, et al. 2017 Mol Biol Evol). We have not elaborated on this further in the revised version of the manuscript because it is beyond the scope of our manuscript.

E. Effects on the mutant selection window are mentioned and documented in a figure. This is an important support for the selection window hypothesis, and that could be mentioned. Also note that the accumulation of resistance alleles raises the mutant selection window has been reported and cited (Li 2004 AAC 48: 4460). There may be other examples.

Our response: We thank the reviewer for bringing the reference to our attention. It is indeed a nice example of how step-wise selection shifts the mutant selection window upwards. We have now included the reference in the revised version of the manuscript, see Line 404.

F. Although English usage is excellent, the excessive use of acronyms slows reading. At every acronym, other than common ones such as DNA, reading stops as

the reader tries to remember the definition. The only reason for using the acronyms is to keep the word count down, but at the cost of reader understanding. The work could also be made more readable by shorter paragraphs. The material is complex, and the reader needs spots to consolidate the information. More “Thus” sentences would help the flow of the story. Comma errors and hyphenation errors with compound adjectives suggest that the work could have been more carefully prepared.

Our response: We agree that our use of acronyms was excessive. These have been considerably reduced, the manuscript was thoroughly proofread, and we have introduced more paragraph breaks to facilitate readability. All changes are highlighted in the revised version.

SPECIFIC, OFTEN MINOR COMMENTS (some may be redundant with the above; items with asterisk are important to consider)

Line 50: I would like to see a reference for the statement that the rate is lagging.

Our response: Reference added on line 52 (Davies, 2006 Can J Inf Dis Med Microbiol).

Line 105: should you mention that the single mutants were, in many cases, obtained by repeated challenge to clarify the method?

Our response: We have now clarified this further in the methods section, which now reads: “Briefly, resistant mutants were selected on Mueller Hinton II agar (MHA-SA; Sigma-Aldrich) step-wise with increasing concentrations of ciprofloxacin, nitrofurantoin and trimethoprim until there was growth above the clinical breakpoint (63).” Further we state that mecillinam mutants were selected in a single step. “Mecillinam resistant mutants and first-step ciprofloxacin mutants (gyrA mutation-containing) were selected as single-step mutants on LB or MHA-SA agar, respectively.” Additionally this was also described within the results section, lines 112-114 and 195-198.

*Line 114: please explain why you used IC-90 rather than IC-50. I would expect plots of IC-50, due to the shape of the curve, to be more sensitive to small differences than IC-90. There is a large, old literature dealing with this issue and use of all data points to estimate IC-50. Since you probably have the data, you may want to reconsider your assay and perhaps consider probit analysis. While this issue may not invalidate the overall conclusions, it seems that you could be more rigorous, which would improve author credibility.

Our response: Please refer to our response to point B above concerning what is frequently used in the existing literature and the linear relationship between IC90 and IC50 as described by Munch et al. 2014 Sci Transl Med. We have

considered the probit analysis as suggested, but we are confident that our analysis is robust and this is not needed.

Line 121 networks. Are you sure this is the right/best word? For most readers networks will mean sets of interacting genes that are identified by the work. You may be misleading the reader here.

Our response: We and other authors in the field use the term “network” to indicate the linkages between responses to different antibiotics, and although we understand that “network” can mean different things in other contexts, our use here is consistent with the terminology in the literature (Imamovic, L, Sommer, MO. 2013 Sci Transl Med; Lazar, V, Pal Singh, G, et al. 2013 Mol Syst Biol). We have added these references in the manuscript at our first use of this term, see line 82.

Line 132 changes does not mean only reduced – changes could mean increased.

Our response: Thank you for this point. This has now been clarified in the revised manuscript, see lines 138-139.

Line 156 clinically relevant. This term is vague, since the changes could influence the emergence of resistance (rather than cure rate), which would be clinically relevant.

Our response: This point is well taken; this section is now part of the legend for Supplementary Fig. 2 in SI, and now reads “Collateral responses were most frequent among ciprofloxacin resistant mutants, where IC_{90} values were above clinical breakpoints for six mutants to chloramphenicol and in K56-68 CIP^R to amoxicillin (a).”

Line 158. Do you offer an explanation for the unusual behavior of ciprofloxacin-resistant mutants? I would minimize thinking on the part of the reader.

Our response: We have addressed this in a new paragraph in the discussion on efflux pumps (see lines 328-336), also see our response to point C above.

Line 164 compound adjective requires hyphen

Our response: Corrected, see line 171.

Line 166 resistance. I think you mean susceptibility, since you are not dealing with breakpoints here (breakpoints are key to the definition of resistance)

Our response: Corrected, see line 173.

Line 190. These data seem to be largely expected from previous work. Do you want

to say “As expected”? I think it is good to distinguish what is really new in the present work from what is basically confirmatory.

Our response: *This point is well taken; this section now reads: “As expected trimethoprim resistant mutants had few collateral responses, likely due to the specific mechanism of resistance affecting a single unique drug target...”. See line 203.*

Line 201 pan-susceptible by growth criteria. But you are looking at the emergence of additional alleles, and there are reports in the literature that par mutations can dramatically increase the selection of quinolone resistant gyrA alleles (see Zhao PNAS 1998). Thus, you need to rethink your statement here.

Our response: *Thank you for this point. We have now modified this statement to read: “Though all of the wild-type strains were phenotypically pan-susceptible (Supplementary Fig. 1c), these resistance determinants could affect antimicrobial susceptibilities differentially in the presence of other mutations (8-10).” See whole genome sequencing section of Supplementary Methods. We have also included references that report mutations in parE that accumulate together with mutations described in our study, leading to elevated ciprofloxacin MICs.*

Line 225: limited is vague. Do you mean minor? Non-significant – do you mean insignificant? Or do you mean not significant with respect to a statistical parameter?

Our response: *We have modified this statement and added a p-value so that is clear that it is not statistically significant, see line 242.*

Line 226. Is your data really accurate to three significant figures? Also, the meaning of this number requires comparison with results from mutations (reiterate if stated above)

Our response: *We originally provided the percent explained variation (% of total inertia) rounded to two decimals in the original submission (main text). We have now further adjusted figures to a single decimal throughout the manuscript, except for p-values which are provided as calculated by the permutation tests.*

Comparison to mutations: *A limitation of our study, from a biological perspective, is that it is difficult to fully disentangle the individual effects of strain background and resistance mutations. It is for this reason that we used the statistical modeling approach to estimate the contribution of each individual factor in order to help explain the phenotypic variation.*

Line 254 and elsewhere you need to define significant, perhaps with p values.

Our response: In the initial version of the manuscript we chose to add the significance threshold in the methods section, where we explicitly state it to be $p < 0.05$. To ensure readability we have decided to keep this original statement (see line 515) and all p-values are available in Fig. 3, supplemental figures S5a-S5k, and Supplementary Table 4. However, in response to the comment above, we make sure to reserve the term “significant” to refer to statistical significance.

Line 281. How did you demonstrate that the isolates were genetically diverse? What is your definition of genetically diverse? One line 387 you fail to specify. Is diversity expected because the isolates were from different countries? That seems like a weak argument. This is not a trivial point, since even with whole genome sequencing you have a spectrum. Thus, you will need some definition. Or rephrase genetic diversity to lack definition and still try to maintain credibility.

Our response: This point is well taken, and also brought up by the other two reviewers. In the revision we include a phylogenetic tree based on whole genome sequencing data (specifically the 3.2 Mbp core genomes) to better describe the genetic diversity and relationships between these strains. This approach improves the resolution provided by our previous approach using MLST. This is now presented in Supplementary Fig. 1a-b and the methods (in Supplementary Methods) have been updated accordingly. Based on pair-wise comparison of the complete genomes, the closest and most distantly related strains varied by approximately 7,200 and 105,000 SNPs, respectively.

Line 375: unknown. Since you use the word action, you are covering both bacteriostatic and bactericidal effects. The word unknown is misleading with respect to lethal action, largely because growth rate, in particular stationary phase, can reduce the production of reactive oxygen species which are now well-established as contributors to antimicrobial-mediated killing. A recent reference on growth phase is Collins Mol Cell for mechanism for a phenomenon that dates back at least to the 1960s.

Our response: We thank the reviewer for bringing this our attention. This sentence was ambiguous. The sentence now reads “We used relative growth rates as a proxy for relative fitness and our data are consistent with reports demonstrating that growth rates affect susceptibilities to several antimicrobials (56, 57). ” (See lines 434-436).

Table S1. Single replicate. If the data used in the analysis derived from multiple replicates, mention that. Also, spelling of drugs in footnotes.

Our response: These data were based on single MIC gradient strip diffusion assays with validated control strains as an assay control (as is the standard routine in medical microbiology laboratories). We have proof-read the text and

corrected all spelling errors.

*Table S4. DNA mutations. Is there another kind? I cannot believe that your data are significant in some cases to 6 significant figures. These numbers suggest a lack of understanding of significant figures and undermines author credibility.

Our response: If Reviewer 1 refers to the inertia columns in Supplementary Table 4, these are output values from our computations, provided here in the supplementary information as raw data, and as such do not want to modify it as others may use it for confirmatory and further analysis.

Line 109 You might want to mention that in some cases multiple rounds of selection were used.

Our response: This is addressed as indicated above (See response to reviewer's point for line 105, above).

*Fig 4. The situation portrayed here, drug concentrations in patients exceeding MPC, has been reported for only a few drug-pathogen combinations (one is ciprofloxacin vs *H. influenzae*; another is moxifloxacin vs *S. pneumoniae*). In general, C_{max} is inside the selection window. The authors should be careful, perhaps by redrawing the figure or by modifying the legend, to avoid misleading the reader.

Our response: In the revised version of the manuscript we make it clear in the Figure legend that this is a schematic illustration and that the exact scenario illustrated does not represent the majority of drug-pathogen combinations. See legend for Fig. 4a-b.

Reviewer #2 (Remarks to the Author):

The manuscript by Podnecky et al describes the collateral sensitivity networks of the *E. coli* strains from UTI. Single isolates for each drug resistant populations were tested against the panel of 16 antibiotics to determine the change in susceptibility. Resistance mechanisms and mutant selection windows were further investigated as well.

Major comments

Note from authors: We have responded within the original paragraph to more easily address aggregated comments.

The study expands the number of *E. coli* isolates tested from 3 strains (Imamovic and Sommer, 2013) to 10 clinical strains in the current study. While the research performed by Podnecky and colleagues expands to broader collection of isolates then used before for *E. coli*, the novelty of this study is somewhat limited.

Our response: We respectfully disagree. As pointed out in our manuscript, and recognized by two other expert reviewers, we demonstrate for the first time that strain background contributes less to collateral responses than both specific resistance mechanisms and the fitness cost of resistance. We also show that even small collateral changes lead to shifts in the mutant selection window with potential implications for resistance evolution in multi-drug environments. Moreover, our data suggest that rapid identification of resistance mechanisms is key for future application of CS and CR clinically.

The seminal paper referred to by reviewer 2 from the Sommer group included two (not 3) clinical isolates that were added to support these authors' findings in *E. coli* MG1655. We would also like to point out that the initial 10 strains were the basis for a total of 40 (now 49) isolates with reduced susceptibility to antimicrobials that are well characterized in this study.

The major driver for the study was said to be not explored direction for genetically diverse strains. However, the authors only use 10 strains as well, this small collection of isolates it is only limited to pan-susceptible strains as well. The authors could offer further information for strain selection. How diverse these strains are (e.g. number of SNPs)?

In particular the term used "in depth study" across genetically diverse isolates seems over statement.

Our response: This point is well taken, and also brought up by the other two reviewers. In the revision we include a phylogenetic tree based on whole genome sequencing data (specifically the 3.2 Mbp core genomes) to better describe the genetic diversity and relationships between these strains. This approach improves the resolution provided by our previous approach using MLST. This is now presented in Supplementary Fig. 1a-b and the methods (in

Supplementary Methods) have been updated accordingly. Based on pair-wise comparison of the complete genomes, the closest and most distantly related strains varied by approximately 7,200 and 105,000 SNPs, respectively.

We have also removed the term “in depth” from the mentioned sentence, see line 319.

All isolates were susceptible to antibiotics and such scenario does lead to the misleading conclusion no matter what the genetic background is, it would lead to conserved responses (as also recognised by authors in the discussion section).

Our response: The pan-susceptibility of the initial strain collection is a major strength of this study because it provides a strong base-line control to detect small changes in relative susceptibility (i.e. small changes in collateral susceptibilities). There is no necessary reason that this would lead to conserved responses independent of strain genetic background. Indeed, our results clearly show that only 19 of 60 collateral responses were conserved (Figs 1 and S2 taken together).

Minor comments

Line 299. E. coli MG1655 is the E. coli strain as well. Please specify the strain differences correctly from the appropriate references quoted, if any.

Our response: Both strains are indeed E. coli. We here revised the text to eliminate any confusion, see line 347-348.

Reviewer #3 (Remarks to the Author):

The topic of the paper is very timely and important; collateral-sensitivity (CS) informed therapy is a promising selection inversion strategy to combat antimicrobial resistance. With this work the authors intend to fill the research gap regarding the conservation of CS networks across diverse clinical E. coli strains. To this purpose i) they chose ten different UTI clinical E. coli isolates, ii) generated antimicrobial resistant mutants from each of these isolates with resistance levels above clinical breakpoints against four different antibiotics used in UTI treatment and iii) they investigated the collateral networks of the resistant mutants against 16 antimicrobials from different drug classes.

By performing a multivariate statistical analysis they found that genetic background did not contribute significantly to collateral responses and based on this they claim that collateral responses are conserved across different E. coli clinical strains. In addition to this their main conclusion is that resistance mechanisms, especially efflux related mutations are the key predictors of collateral responses.

Major concerns:

Conservation of the collateral responses across diverse clinical E. coli strains is the main result of this paper. To be able to draw this important conclusion the authors need to present a figure with the phylogenetic tree of E. coli strains where they place the ten studied E. coli isolates and show that indeed these are different from each other. This is indispensable to draw a conclusion about conservation.

Our response: We appreciate this comment and agree that it is an important point that was also brought up by the other two reviewers. In the revision we include a phylogenetic tree based on whole genome sequencing data (specifically the 3.2 Mbp core genomes) to better describe the genetic diversity and relationships between these strains. This approach improves the resolution provided by our previous approach using MLST. This is now presented in Supplementary Fig. 1a-b and the methods (in Supplementary Methods) have been updated accordingly. Based on pair-wise comparison of the complete genomes, the closest and most distantly related strains varied by approximately 7,200 and 105,000 SNPs, respectively.

Please note also that the manuscript investigates E.coli strains only. It is unknown whether the same results would hold for other enterobacterial species, such as Salmonella.

Our response: We agree that our results do not indicate that CS responses are conserved across species, although this is certainly a very important next-step. We have added this point to the “caveats” section in our discussion, see lines 426-429.

Their second main conclusion is that efflux mutations are the key predictors of collateral responses. However, when they performed the multivariate analysis

separately for each AMR group this conclusion was not valid for trimethoprim and ciprofloxacin resistant mutants. The authors should give a possible explanation for this, especially because almost half of the conserved collateral responses were found in ciprofloxacin resistant mutants.

At the end of the abstract the authors claim that “detection of resistance mechanisms is important to accurately predict collateral antimicrobial responses”, but they do not present a “prediction map” based on their results. Put it simply, they should give some examples that the presence of given mutations in resistant strains would indicate collateral sensitivity or collateral resistance against given antibiotics.

Our response: In a joint response to reviewer 1 and this point we have added a prediction map with suggested secondary treatments as suggested by conserved CS and CR networks. See the modified Fig. 4 c-d. Specifically, we do illustrate the expected collateral responses based on resistance mechanism for ciprofloxacin resistance in Fig. 4c. These new figure panels are discussed on lines 410-418.

Minor concerns:

- Elaborate on the reason of choosing redundancy analysis (RDA) for multivariate statistical analysis.

Our response: A standard principle component analysis is descriptive in that it would simply map the correlations between collateral responses. Our aim was to provide a causal explanation and a quantitative assessment of this variation in susceptibility. Therefore, we chose a multivariate modeling approach to estimate the individual and combined effects of each factor/predictor on the collateral responses. The RDA thus allows us to test specific hypotheses and provide direct quantitative estimates of effects size. We have expanded the explanation for this choice in the text, see lines 504-509.

- In my opinion figure 2 contains too much information making the interpretation difficult. It would be helpful for the reader if the authors could dissect it into separate figures, each highlighting one main conclusion.

Our response: We agree that the complete triplot of the multivariate model contains too much information and have chosen to display a breakdown of those triplots in Fig 3 (formerly Fig 2). Each panel conveys part of the original information in the triplot. Together the two panels provide all the information in the original triplot, as indicated in the text (lines 229-244) and particularly in the figure legend of Fig 3.

In sum, if the authors address the raised concerns in a satisfactory way, I support the publication of the paper in Nature Communications.

REVIEWERS' COMMENTS:

Reviewer #1 (Remarks to the Author):

GENERAL COMMENTS

This paper develops the potentially important idea that resistance to certain antimicrobials can increase the susceptibility to others and thereby affect the choice of compounds during treatment. The work is generally well written, although I have a few minor comments (listed below) that can be easily addressed. I expect the work to impact thinking in the medical community where the focus is on patient cure, not the emergence of resistance. In particular, it offers a way to address resistance that depends on treatment design, not reduction in consumption.

SPECIFIC COMMENTS

Line 38 allow. Do you want to use a stronger word here?

Line 39 early. Meaning is unclear. Do you mean early in the study of a new antimicrobial?

Line 40 frequency. Perhaps better to say prevalence

Line 52 comma after corporate. This is called the Oxford comma and improves writing precision. There are many examples in this paper.

Line 56. awareness of what?

Line 83. few. Do you mean "a few" as in small number or? Or do you mean but few ...?

Line 84 further. Further than what?

Line 90. Change semicolon to colon.

Line 97. different. Do you mean various?

Line 118. Comma but

Line 118. detection not detections

Line 143 and throughout – compound adjectives require a hyphen as in ciprofloxacin-resistant mutants

Line 146 comma and (this is a compound sentence). The absence of a comma in a compound sentence slows reading. This is a problem throughout the text.

Table 1 needs clarification to be independently understandable. The title says selected in vitro. Where do you indicate which drug was used to select which mutant? Or is the strain number the wild type and these data represent mutants selected to each drug? This can be clarified with a footnote.

Line 185. gyrA and efflux. I think you mean both. If so, say both gyrA and efflux

Line 195 delete were?

Line 199 comma after total

Line 203 comma after expected

Lines 334 and 335. Your change in nomenclature, from resistance to loss of susceptibility, create a reading pause and the question why you did this.

Line 411 few collateral responses. These can be both CR and CS, but I don't think you mean both here

Line 480 first. Do you mean lowest?

Line 483 average. Was there much variation among replicates?

Line 492 pelleted is lab jargon and imprecise

Line 498. With such large inocula, colonies can arise from undefined "inoculum effects." Thus, it is prudent to retest colonies to assure that they contain a resistance mutation. This is generally not a problem with E. coli and quinolones.

Karl Drlica

Reviewer #2 (Remarks to the Author):

The authors have addressed the points raised by the reviewers and the manuscript seems technically sound. However, overall this reviewer still considers the manuscript a limited advance to the collateral sensitivity / resistance research area.

Reviewer #3 (Remarks to the Author):

The authors adequately addressed my concerns in revised the manuscript: I have no further comments and criticism.

Response to reviewer 1: SPECIFIC COMMENTS:

Line 38 allow. Do you want to use a stronger word here?

Line 39 early. Meaning is unclear. Do you mean early in the study of a new antimicrobial?

Our responses: We decided to keep “allow”, but we agree that early is imprecise and have replaced it with “rapid”. See Line 42.

Line 40 frequency. Perhaps better to say prevalence

Our response: We agree that prevalence is a better/more accurate term and have corrected this in Line 67.

Line 52 comma after corporate. This is called the Oxford comma and improves writing precision. There are many examples in this paper.

Our response: The sentence has been edited according to the reviewer’s suggestion, see Line 72. This has now been consistently amended throughout the text.

Line 56. awareness of what?

Our response: The sentence has been edited to increase the precision, see Lines 76-77.

Line 83. few. Do you mean “a few” as in small number or? Or do you mean but few ...?

Our response: The sentence has been edited to increase precision, see Line 111.

Line 84 further. Further than what?

Our response: “Further” has been removed from this sentence, see Line 112.

Line 90. Change semicolon to colon.

Our response: The sentence has been edited according to the reviewer’s suggestion, see Line 120.

Line 97. different. Do you mean various?

Our response: As suggested, “different” has been changed to “various”, see Line 135.

Line 118. Comma but

Our response: The sentence has been edited according to the reviewer’s suggestion, see Line 156.

Line 118. detection not detections

Our response: The sentence has been edited according to the reviewer's suggestion, see Line 156.

Line 143 and throughout – compound adjectives require a hyphen as in ciprofloxacin-resistant mutants

Our response: This has now been consistently amended throughout the text.

Line 146 comma and (this is a compound sentence). The absence of a comma in a compound sentence slows reading. This is a problem throughout the text.

Our response: This has been edited according to the reviewer's suggestion, see Line 195. This has now been consistently amended throughout the text.

Table 1 needs clarification to be independently understandable. The title says selected in vitro. Where do you indicate which drug was used to select which mutant? Or is the strain number the wild type and these data represent mutants selected to each drug? This can be clarified with a footnote.

Our response: We thank the reviewer for this comment, and we have tried to clarify this in the footnotes.

Line 185. gyrA and efflux. I think you mean both. If so, say both gyrA and efflux

Our response: The sentence has been edited to increase the precision, see Lines 245-249.

Line 195 delete were?

Our response: "Were" has been deleted, see Line 266.

Line 199 comma after total

Our response: A comma was added, see Line 269.

Line 203 comma after expected

Our response: This sentence and the one following were modified to ensure clarity, see Lines 273-278.

Lines 334 and 335. Your change in nomenclature, from resistance to loss of susceptibility, create a reading pause and the question why you did this.

Our response: We agree that this change in wording was unnecessary. This sentence has now been re-written. See lines 526-529.

Line 411 few collateral responses. These can be both CR and CS, but I don't think you mean both here

Our response: Collateral responses can indeed be both CR and CS. Our point is simply that the absence of either effect would maintain the WT susceptibility levels to other antimicrobials. We feel that this is clear in this section of the discussion. Lines 611-613.

Line 480 first. Do you mean lowest?

Our response: The sentence has been edited as suggested, see Line 706.

Line 483 average. Was there much variation among replicates?

Our response: Generally there was very little variation among replicates as long as quality control standards were met.

Line 492 pelleted is lab jargon and imprecise

Our response: The sentence has been revised, see Lines 748-750.

Line 498. With such large inocula, colonies can arise from undefined "inoculum effects." Thus, it is prudent to retest colonies to assure that they contain a resistance mutation. This is generally not a problem with E. coli and quinolones.

Our response: Additional information has been added (moved from the Supplementary Methods, as requested by the Editor) on our approach to avoid the contribution of inoculum effects to our results. See Lines 753-754.